# BIG Modulates Stem Cell Niche and Meristem Development via SCR/SHR Pathway in Arabidopsis Roots

**DOI:** 10.3390/ijms23126784

**Published:** 2022-06-17

**Authors:** Zhongming Liu, Ruo-Xi Zhang, Wen Duan, Baoping Xue, Xinyue Pan, Shuangchen Li, Peng Sun, Limin Pi, Yun-Kuan Liang

**Affiliations:** 1State Key Laboratory of Hybrid Rice, Department of Plant Sciences, College of Life Sciences, Wuhan University, Wuhan 430072, China; 2017102040063@whu.edu.cn (Z.L.); zhangruoxi@wbgcas.cn (R.-X.Z.); duanwen@whu.edu.cn (W.D.); xuebaoping@whu.edu.cn (B.X.); 2018202040084@whu.edu.cn (X.P.); lishuangchen2012@163.com (S.L.); 00033070@whu.edu.cn (P.S.); 2Hubei Hongshan Laboratory, Wuhan 430070, China; 3State Key Laboratory of Hybrid Rice, The Institute for Advanced Studies, Wuhan University, Wuhan 430072, China

**Keywords:** BIG, stem cell niche, polar auxin transport, SHORT ROOT, SCARECROW, PLETHORA

## Abstract

BIG, a regulator of polar auxin transport, is necessary to regulate the growth and development of Arabidopsis. Although mutations in the *BIG* gene cause severe root developmental defects, the exact mechanism remains unclear. Here, we report that disruption of the *BIG* gene resulted in decreased quiescent center (QC) activity and columella cell numbers, which was accompanied by the downregulation of *WUSCHEL-RELATED HOMEOBOX5* (*WOX5*) gene expression. BIG affected auxin distribution by regulating the expression of PIN-FORMED proteins (PINs), but the root morphological defects of *big* mutants could not be rescued solely by increasing auxin transport. Although the loss of *BIG* gene function resulted in decreased expression of the *PLT1* and *PLT2* genes, genetic interaction assays indicate that this is not the main reason for the root morphological defects of *big* mutants. Furthermore, genetic interaction assays suggest that BIG affects the stem cell niche (SCN) activity through the SCRSCARECROW (SCR)/SHORT ROOT (SHR) pathway and BIG disruption reduces the expression of *SCR* and *SHR* genes. In conclusion, our findings reveal that the *BIG* gene maintains root meristem activity and SCN integrity mainly through the SCR/SHR pathway.

## 1. Introduction

In Arabidopsis (*Arabidopsis thaliana* L.), the indeterminate growth of the root is supported by a series of undifferentiated stem cells, which are located in the root apical meristem (RAM). The RAM pattern is formed by a range of asymmetric divisions of stem cells around the quiescent center (QC) that is almost undivided under normal conditions [1,2,3]. These stem cells, together with the QC in contact with them, make up the stem cell niche (SCN), which supplies the source of cells for RAM [1,2,3]. Post-embryonically, the expression of the *WUSCHEL-RELATED HOMEOBOX5* (*WOX5*) gene is strictly limited to QC for defining QC and maintaining the undifferentiated condition of columella stem cells (CSCs) [4,5]. Loss of *WOX5* gene function results in the differentiation of CSCs that displays the accumulation of starch, which is also accompanied by an additional division of the QC [4,5].

The SCRSCARECROW (SCR)/SHORT ROOT (SHR) and PLETHORA (PLT) pathways are two independent pathways involved in the regulation of SCN activity and root meristem size [6]. Both SHR and SCR are members of the GRAS transcription factor family that are necessary for the maintenance of SCN activity [6,7]. SHR is synthesized in the stele and then transported to the neighboring cell layer [8], which nucleates the assembly of the SHR/SCR heterocomplex [9,10,11,12,13]; this heterocomplex promotes SHR nuclear localization and increases SCR expression, which in turn constrains the movement of the SHR protein to the next layer of cells [8,11]. Both *shr* and *scr* mutants display defective QC maintenance and fail to maintain root meristem activity that leads to a phenotype of shorter meristems [8,11,14]. Mutations in the *SCR* and *SHR* genes cause the cortex/endodermis initially to fail to complete normal asymmetric division, resulting in the formation of only one layer of ground tissue [9,10,11,12,13].

Auxin and its gradient distribution play important roles in a wide range of growth and development processes, including embryogenesis, organogenesis, cell determination and division, tissue patterning, and tropisms in plants [15,16,17,18]. Additionally, auxin controls proper root pattern formation and SCN maintenance [16,19]. The directional transport of auxin allows for the asymmetric distribution of auxin in distinct cells and tissues [15] to generate local auxin maxima, minima, and gradients for organ initiation and shape determination [16,17]. The auxin-inducible *PLETHORA* (*PLT1* and *PLT2*) genes, which encode two members of the AP2 class transcription factors, are required for the transcription of numerous root development-related genes, including genes that are involved in auxin biosynthesis and transport [20,21,22]. *plt1 plt2* double mutants cannot resolve correct QC morphology and show a dramatic reduction in root meristem length [20,23]. Interestingly, loss of PLT1 and PLT2 function did not affect SCR and SHR expression, although *plt1 plt2* mutants, *scr*, and *shr* mutants all exhibited SCN defects [20], suggesting that the PLTs pathway and the SHR/SCR pathway are parallel and converge to maintain the SCN [20,21,22].

Ubiquitin protein ligase E3 component N-recognin 4 (UBR4) is a 600-kDa calmodulin-interacting protein involved in the N-end rule pathway [24,25,26,27]. BIG was originally identified in a mutational screen for resistance against N-1-naphthylphthalamic acid (NPA), a potent auxin transport inhibitor [28]. BIG is the only UBR4 homolog in Arabidopsis [29], which has been demonstrated to modulate circadian adjustment [30] and C/N balance [31]. Mutations in the *BIG* gene result in multiple defects, including altered aerial organ development, impaired hormone and light signaling, root growth and lateral root development, and aberrant stomatal CO_2_ responses and immunity [28,32,33,34,35,36,37,38].

Previous work has shown that the *BIG* gene is involved in regulating polar auxin transport and *big* mutants show severe root developmental defects [28,29,35]. In this study, we sought to develop a better understanding of how the *BIG* gene is involved in the root development and found that the *BIG* gene is integral to the maintenance of SCN and meristem activity, and this function depends on the SCR/SHR pathway. Mutations in the *BIG* gene lead to a severe reduction in meristem activity and reduced QC activity. Genetic analysis shows that the *WOX5* gene has epistatic effects on the *BIG* gene in maintaining SCN. Decreased levels of PIN-FORMED proteins (PINs) are likely a part of the reasons for the auxin transport defects in *big* mutants, though increasing auxin transport could not rescue phenotypic defects in *big* mutant roots. Loss of BIG function results in the downregulation of *PLTs* gene expression, whereas the *big* mutant has additive effects with the *plt1-4 plt2-2* mutants on the control of root growth. Further genetic experiments show that BIG regulates root development in an SCR/SHR-dependent manner. Consistently, the expression of *SCR* and *SHR* genes was reduced in the *big* mutants. Taken together, we propose that the *BIG* gene participates in maintaining SCN and meristem activity through the SCR/SHR pathway.

## 2. Results

### 2.1. The BIG Gene Is Involved in the Maintenance of SCN Integrity

As evident from previous work, *big* mutants showed a much-reduced root length compared to the wild-type (WT, Col-0) seedlings at 5 days after germination (DAG) [28,29,30,31,32,33,34,35]. We questioned whether BIG participates in the maintenance of SCN. The expression of *QC46* [5], which is a QC-specific marker, was used to measure QC activity in WT and *big-1*. The slightly decreased expression of *QC46* in *big-1* indicates that BIG participates in maintaining the activity of the QC (Figure 1A). Endoplasmic reticulum (ER)-localized GFP driven by the *WOX5* promoter (*pWOX5::erGFP*) [39] were crossed into *big* mutants before confocal laser scanning microscopy (CLSM) was performed, and the results show that mutations in the *BIG* gene caused a reduction in *WOX5* transcription (Figure 1B,C). In line with this observation, mRNA quantification by the reverse transcription-quantitative PCR (RT-qPCR) revealed a significant downregulation of the *WOX5* expression in *big-1* (Figure 1D).

Given that WOX5 is indispensable for maintaining the CSCs in an undifferentiated state [4,5], we next wanted to know whether the reduction of *WOX5* expression in *big* mutants affects the development of CSCs and the derived columella cells. As shown in Figure 1E,F, the columella cells in *big* mutants had one layer less than those of WT, suggesting that BIG deficiency disturbed the development of CSCs and columella cells. In addition, to verify the genetic relationship between the *WOX5* gene and the *BIG* gene, *wox5-1* [4] was introduced into *big-1*, and the phenotypes of the SCN were monitored. As shown in Figure 1E, the modified pseudo-Schiff propidium iodide (mPS-PI) staining of the *wox5-1 big-1* double mutant showed a similar QC phenotypic defect to *wox5-1* but more severe than that of *big-1*, indicating that the *WOX5* gene has epistatic effects on the *BIG* gene in maintaining SCN. These results indicate that the *BIG* gene is involved in maintaining SCN integrity.

### 2.2. Increasing the Auxin Transport Capacity Cannot Rescue the Phenotypic Defects of Big Mutants

The *BIG* gene is required for regulating the polar auxin transport and the endocytosis, two processes that involve the PIN protein family [28,29,32,34]. To determine the auxin accumulation of *big* mutant in root tips, seedlings expressing GUS under the control of the artificial *DR5* promoter were subjected to examine the concentrations of auxin. As shown in Figure 2A, the GUS staining signals of *pDR5::GUS* [40] in the root tip of *big-1* were significantly less than those of WT, suggesting that fully functional BIG is required for the proper auxin accumulation, in agreement with several previous studies [28,29,32,34]. Given that BIG acts synergistically with PIN1 to control the development of leaves and shoot apical meristems [35], we hypothesized that BIG regulates polar auxin transport by regulating the expression of the PIN proteins. To this end, *big-1* was introduced into the transgenic plants that express translational fusions of PIN:GFP proteins under the control of the native promoter of each individual *PIN* gene. The fluorescence intensity of *pPIN1::PIN1:GFP* [41], *pPIN2::PIN2:GFP* [16], *pPIN3::PIN3:GFP* [16], and *pPIN7::PIN7:GFP* [16] was significantly reduced by a *big-1* mutant (Figure 2C–J), suggesting that BIG influences the polar auxin transport via modulating PINs expression. To further ascertain this observation, RT-qPCR analysis was conducted, which showed great reductions in PINs transcription in *big-1* compared to that of WT (Figure 2B). Together, these findings indicate the capacity of BIG for regulating the expression of PINs family proteins and influencing the polar auxin transport.

Strigolactones modulate not only shoot branching but also the root architecture, such as lateral root formation, root hair elongation, and meristem cell number through a variety of *MORE AXILLARY GROWTH* (*MAX*) genes that positively regulate auxin transport [42,43,44,45]. The *max4-1* mutant was used to efficiently rescue the auxin transport deficiency of *tir3-101* (another allele mutant of the *BIG* gene) but failed to alleviate the over-branched phenotype of *tir3-101* [42]. We wanted to know whether the suppressive effect on auxin transport by *max4-1* could restore the RAM defects in *big* mutants. As shown in Appendix A, the root morphological structure of the *max4-1 tir3-101* double mutant [42] displayed no detectable difference from that of *tir3-101*. Then, the *max4-1 big-1* double mutant expressing *pDR5rev::GFP* [46] was generated to assess auxin concentrations in root tips. As shown in Figure 3A,B, the fluorescence intensity of *pDR5rev::GFP* in *max4-1* was higher than that of WT, but there was no significant difference in the fluorescence intensity between the *max4-1 big-1* double mutant and the WT, suggesting that *max4-1* increased the auxin transport capacity in root tips *in the big-1* mutant. Intriguingly, as shown in Figure 3C–F, compared to *big-1*, the root morphology was not restored in the *max4-1 big-1* double mutant, suggesting that the morphological defects of *big* mutants could not be rescued by manipulating the auxin transport. To further evaluate whether increasing auxin content in *big* mutants could alter the root morphology, *big-1* was crossed with the *YUCCA-OX* transgenic plants that overexpress *YUCCA1*, a key auxin biosynthesis gene [47]. *YUCCA-OX big-1* seedlings had longer hypocotyl and petioles relative to *big-1*. In contrast, *YUCCA-OX big-1* showed no discernable restoration of root morphology compared to *big-1* (Appendix A), suggesting that improving the capacity of auxin transport could not affect the RAM phenotypes in *big* mutants.

### 2.3. BIG and PLT Genes Function Independently in Regulating Root Growth

Auxin-induced gradients of PLTs (PLT1–4) converge at the SCN to form a concentration peak and maintain the stem cell identity [20,21,22,23]. Since restoring the impaired auxin transport capacity could not affect the root phenotypes in the *big* mutant, we hypothesized that BIG deficiency might disturb the auxin-related PLT pathway. The *plt1-4 plt2-2* [20] double mutant was crossed with the *big-1* mutant. Data in Figure 4A–D show that the length of either root or of the RAM in the *plt1-4 plt2-2 big-1* triple mutant was significantly shorter than that in the *plt1-4 plt2-2* double mutant. Moreover, the meristematic cells of the *plt1-4 plt2-2 big-1* triple mutants appeared to be completely differentiated, in marked contrast to the *plt1-4 plt2-2* double mutants (Figure 4C), indicating that *big* mutant has additive effects with *plt1-4 plt2-2* on the control of root growth. Next, we crossed the *big-1* mutant with the transgenic plants that express ER-localized cyan fluorescent protein (CFP) under the control of the *PLT* promoters. The fluorescence intensities of *pPLT1::erCFP* [23] and *pPLT2::erCFP* [23] were lower in *big-1* compared to WT seedlings (Figure 4E–H), indicating that mutation in the *BIG* gene decreased the expression of the *PLT* genes. RT-qPCR analysis of the *PLT* genes lent further support to this outcome (Figure 4I). From this evidence, we conclude that the PLT pathway is not the main cause of the root morphological defects in the *big* mutant.

### 2.4. BIG Contributes to the SHR/SCR Pathway in Regulating Root Growth

Given that the SHR/SCR pathway is parallel to the PLT pathway in regulating root development [20], we then examined the possible relationship between BIG and SHR/SCR. For this purpose, *shr-2* and *scr-3* were crossed with *big-1*, respectively. As shown in Figure 5A–D, the lengths of both root and RAM of *shr-2 big-1* and *scr-3 big-1* double mutants were significantly shorter than *big-1* but apparently similar to that of *shr-2* or *scr-3* single mutant, indicating that *big-1* has no additive effects with *shr-2* or *scr-3* on the root growth. These data suggest that BIG regulates root development via SHR/SCR actions. Next, CLSM was used to observe the root cytological morphology of *shr-2 big-1* and *scr-3 big-1* double mutants. As shown in Figure 5D, the double mutants *shr-2 big-1* and *scr-3 big-1* showed the comparable phenotype of a single GT layer to either *shr-2* or *scr-3* single mutant. Notably, the SCN morphology of either the *shr-2 big-1* or the *scr-3 big-1* double mutants resembles that of *shr-2* or *scr-3* (Figure 5D), indicating BIG likely acts through the SHR/SCR pathway to regulate root patterning and growth. To evaluate whether mutations in the *BIG* gene affect the expression of the *SHR* gene, we crossed *big-1* with the transgenic plants that express erGFP under the control of the *SHR* promoter. The fluorescence intensity of *pSHR::erGFP* [48] in *big-1* was significantly lower than that of WT (Figure 6A,B), suggesting that BIG positively regulates the expression of the *SHR* gene. Likewise, to determine whether a *big* mutant affects *SCR* expression, we crossed the *big-1* mutant with *pSCR::GFP* [10]. The fluorescence intensity of *pSCR::GFP* was substantially lower in the endodermis of *big-1* than that in WT (Figure 6C,D), suggesting that BIG positively regulates the expression of *SCR*. RT-qPCR analysis showed that *big* mutant suppresses the expression of the *SHR* and *SCR* genes (Figure 6E). Taken together, these findings indicate that the *BIG* gene is involved in the SHR/SCR pathway that regulates root growth and patterning in Arabidopsis.

## 3. Discussion

Previous studies have shown the *big* mutants with shorter roots and reduced meristem length [28,29,35]. QC maintains the fate of stem cells and prevents them from differentiation, which is important for the development of roots; the QC-specifically expressed *WOX5* gene is indispensable to SCN maintenance [2,4,5]. The *BIG* gene is required to maintain QC activity; the expression of *QC46* and *WOX5* was decreased in the *big* mutant (Figure 1A–D); at the same time, the CSCs development in the *big* mutant was disturbed (Figure 1F). The *wox5-1 big-1* double mutant showed a comparable QC phenotype to that of *wox5-1* (Figure 1E), indicating that *WOX5* has epistatic effects on *BIG* in maintaining SCN. These data indicate that the *BIG* gene is required for maintaining SCN integrity.

The *BIG* gene was originally identified in a genetic screen for mutants that are insensitive to the auxin transport inhibitor NPA and has been assigned a role in polar auxin transport and in auxin-inhibited endocytosis [28,29,32,34]. BIG acts synergistically with PIN1 to control the development of leaves and shoot apical meristems [35,49]. Auxin accumulation in root tips is reduced by a *big-1* mutant (Figure 2A and Figure 3A). Consistently, *BIG* gene deficiency resulted in a marked decrease in the expression of the *PIN* gene family (Figure 2B–J). Our results suggest that BIG impacts polar auxin transport by modulating the expression of the *PIN* genes (Figure 2B–J). However, it is unclear how BIG regulates the expression of the PINs and consequently affects the polar transport of auxin in root tips. A possible explanation is that BIG regulates the expression of PINs through auxin-inhibited endocytosis, which in turn modulates the polar transport of auxin [34]. The *max4-1* mutant has been used to efficiently increase the auxin transport activity but failed to alleviate the over-branched phenotype of the *big* mutant [42]. Neither increasing auxin transport capacity nor by increasing the auxin levels in *big* mutants could restore the RAM defects (Figure 3A–F and Appendix A), suggesting that the impaired auxin transport capacity is not, at least not mainly, responsible for the root developmental defects in *big* mutants, in line with previous reports that mutations in the *BIG* gene hardly caused any abnormal responses to auxin applications [28,35].

Auxin and PLTs function together to render the feedback regulation in the root development system [20,21,22,23]. *plt1-4 plt2-2* double mutants exhibit prematurely differentiated root meristems [20]. This study shows that the meristem exhaustion in the *plt1-4 plt2-2* double mutant is faster than that in the *big* mutant background (Figure 4A–D), which suggests that BIG functions without relying on the PLT pathway in regulating the RAM activity and SCN integrity. However, disruption of BIG could suppress the expression of the *PLTs* genes (Figure 4E–I). This is likely due to the relatively lower auxin accumulation in the enlarged roots caused by the *big* mutant, which subsequently suppressed the auxin-induced *PLTs* expression (Figure 2A and Figure 3A). Together, these findings strongly indicate that despite its involvement in auxin transport, BIG functions through a pathway independent of the PLT pathway in the regulation of root development.

SHR/SCR functions in parallel to the PLTs in regulating root development [20]. Our genetic investigations show that *BIG* deficiency has no additive effects with either *shr-2* or *scr-3* on root growth (Figure 5A–D). Given that mutations in either the *SHR* or *SCR* gene resulted in a single GT layer, whereas *shr’s* GT has cortex cell characteristics and *scr’s* GT has both cortex cell and endodermis cell characteristics, SHR has been ascribed an additional role in the endodermis cell fate specification [9,10,11,12,13]. The double mutants *shr-2 big-1* and *scr-3 big-1* showed similar phenotypes of a single GT layer, indicating that BIG regulates the development of GT via the SHR/SCR pathway (Figure 5D). Our data indicate that the *shr-2 big-1* and *scr-3 big-1* double mutants have similar QC-deficient morphology to either *shr-2* or *scr-3* (Figure 5D, [9,10,11,12,13]). These results suggest that BIG modulates root SCN via SCR/SHR pathway. Disrupting BIG will simultaneously reduce the expression of the *SHR* gene and the *SCR* gene (Figure 6A–E). As the only homolog of pRB in higher plants, RBR is highly conserved across species [50]. To govern cell adhesion in human cells, pRB and UBR4 interact directly and co-localize in the nucleus [24]. UBR4 is a UBR box E3 ligase that uses the N-degron pathway to degrade proteins [24,25,26,27]. Sequence similarity analysis reveals that BIG has the conserved UBR box and E3 ligase domains (Appendix A), pointing to a scenario that BIG might also regulate RBR levels through the N-degron pathway. RBR follows a protein degradation process to abolish its negative regulatory effect on SCR and SHR [50]. Taken together, we propose that BIG modulates meristem development and SCN integrity via the SCR/SHR pathway in Arabidopsis roots and the BIG-mediated polar auxin transport also contributes to this process (Figure 6F). Further investigation is required to experimentally establish a direct relationship between BIG and the SCR/SHR pathway in the future, though, considering the extra-large size of the putative BIG protein, it would be technically challenging.

## 4. Materials and Methods

### 4.1. Plant Material and Growth Conditions

*Arabidopsis thaliana* ecotype Columbia (Col) was used as WT. Seeds of *big-2* (CS903939), *shr-2* (N2972), and *scr-3* (N3997) were obtained from NASC (the European Arabidopsis Stock Centre, http://arabidopsis.org.uk (accessed on January 2018)). Other mutant and transgenic plant lines used in this study including *big-1* [36], *QC46* [5], *pWOX5::erGFP* [39], *wox5-1* [4], *pDR5::GUS* [40], *pPIN1::PIN1:GFP* [41], *pPIN2::PIN2:GFP* [16], *pPIN3::PIN3:GFP* [16], *pPIN7::PIN7:GFP* [16], *pDR5rev::GFP* [43], *max4-1* [42], *tir3-101* [42], *YUCCA1-OX* [47], *plt1-4 plt2-2* [20], *pPLT1::erCFP* [23], *pPLT2::erCFP* [23], *pSHR::erGFP* [48], and *pSCR::GFP* [10] were kind gifts from the authors.

After surface-sterilizing for 10 min in 70% ethanol, the seeds were placed on sterilized filter paper in a laminar air flow hood and blown dry before sowing on a half-strength Murashige and Skoog (1/2 MS, Sigma, St. Louis, MO, USA) media plate [51] with 0.8% agar (Sigma) and 1% sucrose. Next, the seed plates were placed at 4 °C for no less than 2 days in the dark and then vertically placed into a plant greenhouse at 22–23 °C with a 16 h light (light intensity: 120 µmol photons m^−2^s^−1^)/8 h dark photoperiod. If not stated otherwise, Arabidopsis roots were investigated at 5 DAG.

### 4.2. Histology and Microscopy

The GUS histochemical staining was performed as previously described [35] with minor modifications. Briefly, the seedlings were fixed using 90% acetone for 20 min; adding 1 mL of GUS lotion (pH 7.0, 2 mM ferricyanide, 2 mM ferrocyanide, and 10 mM Ethylene Diamine Tetraacetic Acid) to wash twice to remove the acetone; adding GUS staining solution (GUS lotion with 1 mM X-glucuronide), vacuuming on ice for 15–20 min; placing in the dark at 37 °C for an appropriate time according to the experimental conditions; agitating the staining solution, adding 0.8 mL 70% ethanol to stop the staining reaction and decolorization; changing the ethanol several times until the decolorization is complete. An Olympus IX70 microscope (Olympus, Tokyo, Japan) was used to capture the differential interference contrast images and image processing with Olympus cellSens software (1.6, Olympus, Tokyo, Japan).

The mPS-PI staining was performed essentially as previously described [52]. Arabidopsis seedlings were fixed in 2% formaldehyde solution for 25–30 min. The fixed sample was soaked in methanol at room temperature for 15 min, rinsed for 5 min, then placed in a 1% periodic acid solution for 25–30 min at 22–25 °C. Samples were then rinsed for 5 min, treated with Schiff reagent (1 mM sodium thiosulfate, 0.15 M HCl), 5 µg/mL propidium iodide (PI, Sigma) added, and stained for 15–30 min.

For CLSM, the roots of 4–6 DAG Arabidopsis seedlings were incubated in PI (10 μg/mL) for 3–5 min, using a Leica SP8 system (Leica, Wetzlar, German) to observe. Fluorescence of GFP, CFP, YFP, and PI staining was visualized using the settings: excitation wavelength 488 nm and emission wavelength from 505 to 550 nm for GFP, excitation wavelength 458 nm and emission wavelength from 463 to 500 nm for CFP, and excitation wavelength 561 nm and emission wavelength from 600 to 650 nm for PI staining, respectively. Leica AF Lite software (3.3.10, Leica, Wetzlar, German) was used to capture the images. Three biological replicates were generated. We quantified fluorescence intensity using 20–30 roots with ImageJ (1.8.0, National Institutes of Health, USA). The same microscope settings were used to observe WT and *big* seedlings. One-way ANOVA (Tukey’s multiple comparisons test) or Student’s *t*-test was used for significance analysis.

### 4.3. The Reverse Transcription-Quantitative PCR Assays

Approximately 5–10 mm of 5 DAG seedlings’ apical parts were used to extract RNA. cDNA was prepared by PrimeScript™ RT reagent Kit (TaKaRa, Bio Inc. Shiga, Japan) and quantified on a Bio-Rad CFX384 Touch fluorescence quantitative PCR instrument (Bio-Rad, Hercules, CA, USA) with the SYBR^®^ Green Realtime PCR Master Mix (TOYOBO, Tokyo, Japan). The *Actin7* gene was used as an internal reference gene. Three biological replicates were generated. One-way ANOVA (Tukey’s multiple comparisons test) or Student’s *t*-test was used for significance analysis. The primers used are listed in Appendix A.

### 4.4. Accession Number

AT3G02260 (*BIG*), At3g11260 (*WOX5*), AT4G32810 (*MAX4*), AT4G32540 (*YUCCA1*), AT1G73590 (*PIN1*), AT5G57090 (*PIN2*), AT1G70940 (*PIN3*), AT1G23080 (*PIN7*), At3g54220 (*SCR*), AT4G37650 (*SHR*), At3g20840 (*PLT1*), At1g51190 (*PLT2*).

## Figures and Tables

**Figure 1 ijms-23-06784-f001:**
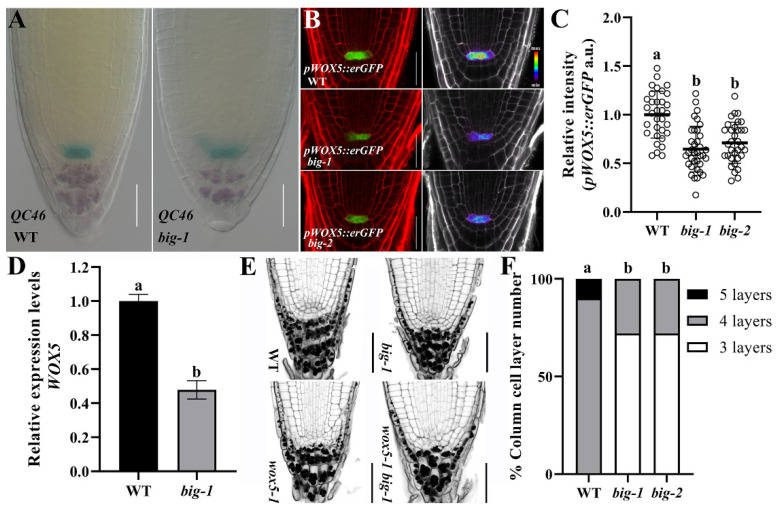
BIG is involved in maintaining the SCN. (**A**) *QC46* expression in WT and *big-1* seedlings at 5 DAG. Scale bars, 20 µm. (**B**) *pWOX5::erGFP* expression in WT and *big* seedlings at 5 DAG. Left, the merge of the PI and GFP channels; right, the GFP channel displayed in pseudo colors with intensity scale. a.u. indicates arbitrary units. Scale bars, 20 µm. (**C**) Quantification of *pWOX5::erGFP* relative intensity in WT and *big* seedlings at 5 DAG. Values are means ± SD (*n* ≥ 30). One-way ANOVA, Tukey’s multiple comparisons test. Different letters indicate values are significantly different (*p* < 0.01). (**D**) qRT-PCR analyses show the relative expression levels of *WOX5* in WT and *big* seedlings. Error bars represent SD. Student’s *t*-test. Different letters indicate values are significantly different (*p* < 0.01). (**E**) mPS-PI staining of WT, *big-1*, *wox5-1*, and *wox5-1 big-1* root tips at 5 DAG. Scale bars, 50 µm. (**F**) Quantification of columella cell layer number in WT and *big* seedlings at 5 DAG. Values are means ± SD (*n* ≥ 30). One-way ANOVA, Tukey’s multiple comparisons test. Different letters indicate values are significantly different (*p* < 0.01).

**Figure 2 ijms-23-06784-f002:**
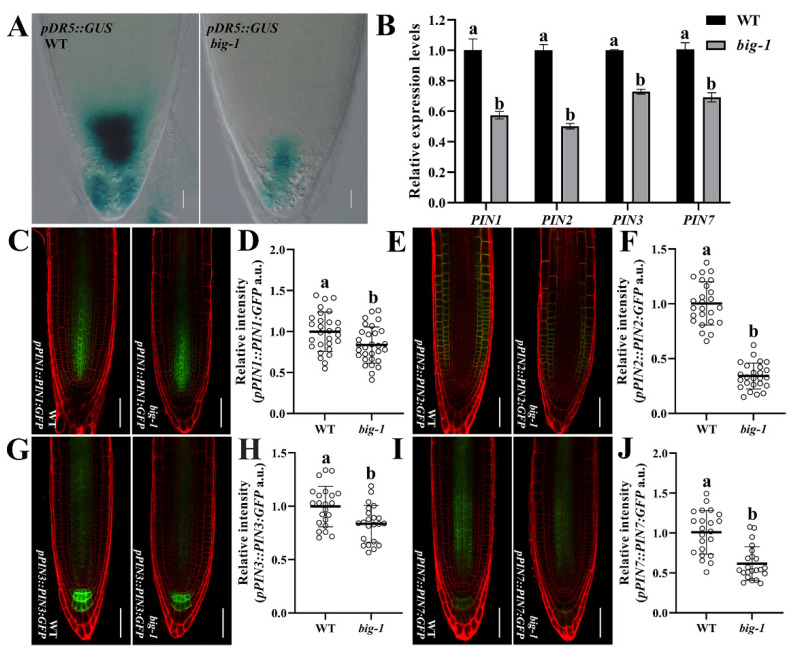
BIG is involved in the regulation of polar auxin transport. (**A**) *pDR5::GUS* expression in WT and *big* seedlings at 5 DAG. Scale bar, 20 µm. (**B**) qRT-PCR analyses show the relative expression levels of *PIN1*, *PIN2*, *PIN3*, and *PIN7* in WT and *big-1* seedlings. Error bars represent SD. Two-way ANOVA, Tukey’s multiple comparisons test. Different letters indicate values are significantly different (*p* < 0.05). (**C**,**E**,**G**,**I**) *pPIN1::PIN1:GFP* (**C**), *pPIN2::PIN2:GFP* (**E**), *pPIN3::PIN3:GFP* (**G**), and *pPIN7::PIN7:GFP* (**I**) expression in WT and *big-1* seedlings at 5 DAG. Scale bar, 50 µm. (**D**,**F**,**H**,**J**) Quantification of *pPIN1::PIN1:GFP* (**D**), *pPIN2::PIN2:GFP* (**F**), *pPIN3::PIN3:GFP* (**H**), and *pPIN7::PIN7:GFP* (**J**) relative intensity in WT and *big-1* seedlings at 5 DAG. Values are means ± SD (*n* ≥ 30). Student’s *t*-test. Different letters indicate values are significantly different (*p* < 0.01).

**Figure 3 ijms-23-06784-f003:**
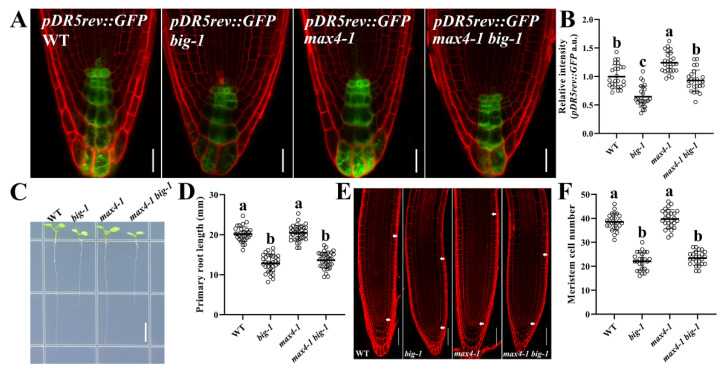
Increasing auxin could not restore the root morphology in *big* mutants. (**A**) *pDR5rev::GFP* expression in WT and *big* seedlings at 5 DAG. Scale bar, 16 µm. (**B**) Quantification of *pDR5rev::GFP* relative intensity in WT, *big-1*, *max4-1*, and *max4-1 big-1* seedlings at 5 DAG. Values are means ± SD (*n* ≥ 25). One-way ANOVA, Tukey’s multiple comparisons test. Different letters indicate values are significantly different (*p* < 0.01). (**C**) Images of the indicated genotypes WT, *big-1*, *max4-1*, and *max4-1 big-1* at 5 DAG. Scale bar, 5 mm. (**D**) Primary root length of WT, *big-1*, *max4-1*, and *max4-1 big-1* at 5 DAG, Values are means ± SD (*n* ≥ 30). One-way ANOVA, Tukey’s multiple comparisons test. Different letters indicate values are significantly different (*p* < 0.01). (**E**) Representative CLSM image of WT, *big-1*, *max4-1*, and *max4-1 big-1* at 5 DAG. Scale bar, 50 µm. (**F**) Quantification of meristem cell number of WT, *big-1*, *max4-1*, and *max4-1 big-1* seedlings at 5 DAG. Values are means ± SD (*n* ≥ 25). One-way ANOVA, Tukey’s multiple comparisons test. Different letters indicate values are significantly different (*p* < 0.01).

**Figure 4 ijms-23-06784-f004:**
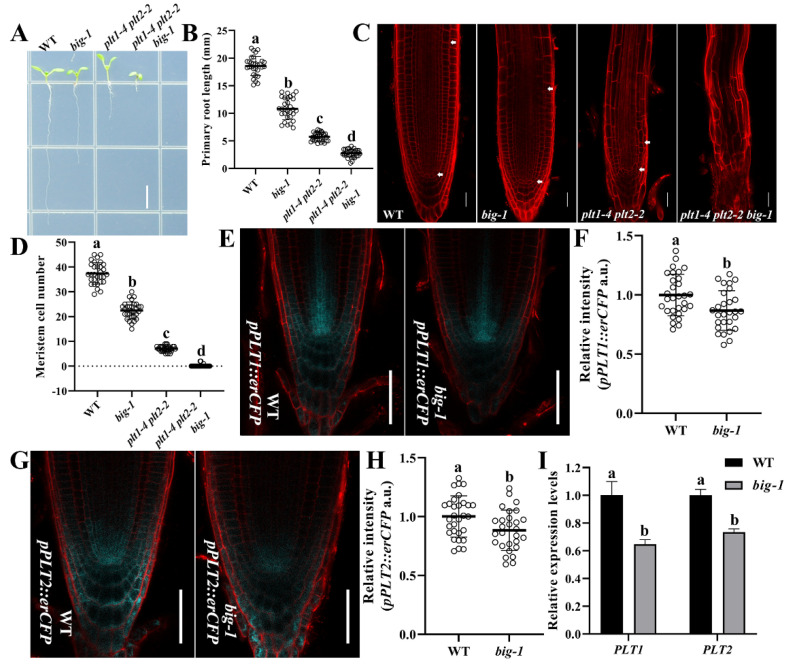
BIG and PLTs function in the independent pathway. (**A**) Images of the indicated genotypes WT, *big-1*, *plt1-4 plt2-2*, and *plt1-4 plt2-2 big-1* seedlings at 5 DAG. Scale bar, 5 mm. (**B**) Primary root length of WT, *big-1*, *plt1-4 plt2-2*, and *plt1-4 plt2-2 big-1* seedlings at 5 DAG. Values are means ± SD (*n* ≥ 30). One-way ANOVA, Tukey’s multiple comparisons test. Different letters indicate values are significantly different (*p* < 0.01). (**C**) Representative CLSM image of WT, *big-1*, *plt1-4 plt2-2*, and *plt1-4 plt2-2 big-1* seedlings at 5 DAG. Scale bar, 25 µm. (**D**) Quantification of meristem cell number of WT, *big-1*, *plt1-4 plt2-2*, and *plt1-4 plt2-2 big-1* seedlings at 5 DAG. Values are means ± SD (*n* ≥ 30). One-way ANOVA, Tukey’s multiple comparisons test. Different letters indicate values are significantly different (*p* < 0.01). (**E**,**G**) *pPLT1::erCFP* (**E**) and *pPLT2::erCFP* (**G**) expression in WT and *big-1* seedlings at 5 DAG. Scale bars, 20 µm. (**F**,**H**) Quantification of *pPLT1::erCFP* (**F**) and *pPLT2::erCFP* (**H**) relative intensity in WT and *big-1* seedlings at 5 DAG. Values are means ± SD (*n* ≥ 20). Student’s *t*-test. Different letters indicate values are significantly different (*p* < 0.05). (**I**) qRT-PCR analyses show the relative expression levels of *PLT1* and *PLT2* in WT and *big-1* seedlings. Error bars represent SD. Two-way ANOVA, Tukey’s multiple comparisons test. Different letters indicate values are significantly different (*p* < 0.01).

**Figure 5 ijms-23-06784-f005:**
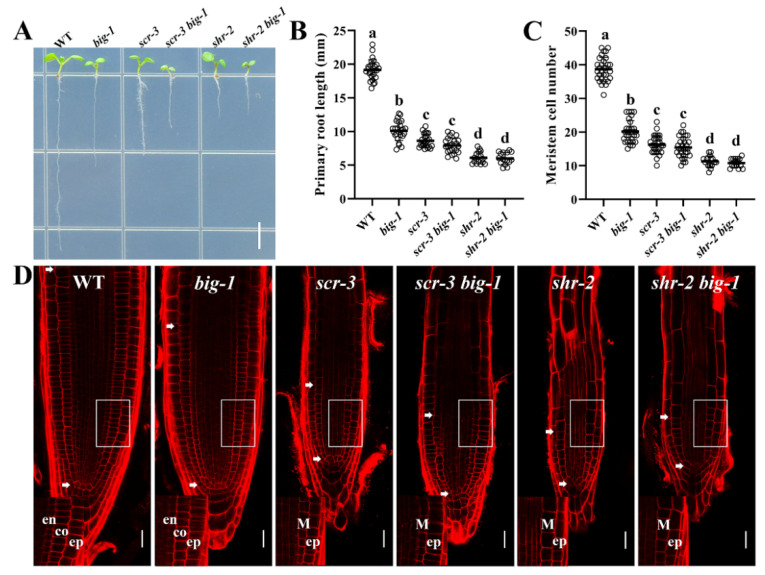
BIG acts in the SHR/SCR pathway to regulate root growth. (**A**) Images of the indicated genotypes WT, *big-1*, *scr-3*, *scr-3 big-1*, *shr-2*, and *shr-2 big-1* seedlings at 5 DAG. Scale bar, 5 mm. (**B**) Primary root length of WT, *big-1*, *scr-3*, *scr-3 big-1*, *shr-2*, and *shr-2 big-1* seedlings at 5 DAG. Values are means ± SD (*n* ≥ 30). One-way ANOVA, Tukey’s multiple comparisons test. Different letters indicate values are significantly different (*p* < 0.01). (**C**) Quantification of meristem cell number of WT, *big-1*, *scr-3*, *scr-3 big-1*, *shr-2*, and *shr-2 big-1* seedlings at 5 DAG. Values are means ± SD (*n* ≥ 30). One-way ANOVA, Tukey’s multiple comparisons test. Different letters indicate values are significantly different (*p* < 0.01). (**D**) Root apical phenotypes of WT, *big-1*, *scr-3*, *scr-3 big-1*, *shr-2*, and *shr-2 big-1* at 5 DAG. The insets show root radial patterning surrounded by white rectangles. en—endodermis; co—cortex; ep—epidermis; m—mutant cell layer in *shr-2* or *scr-3*. Scale bars, 50 µm.

**Figure 6 ijms-23-06784-f006:**
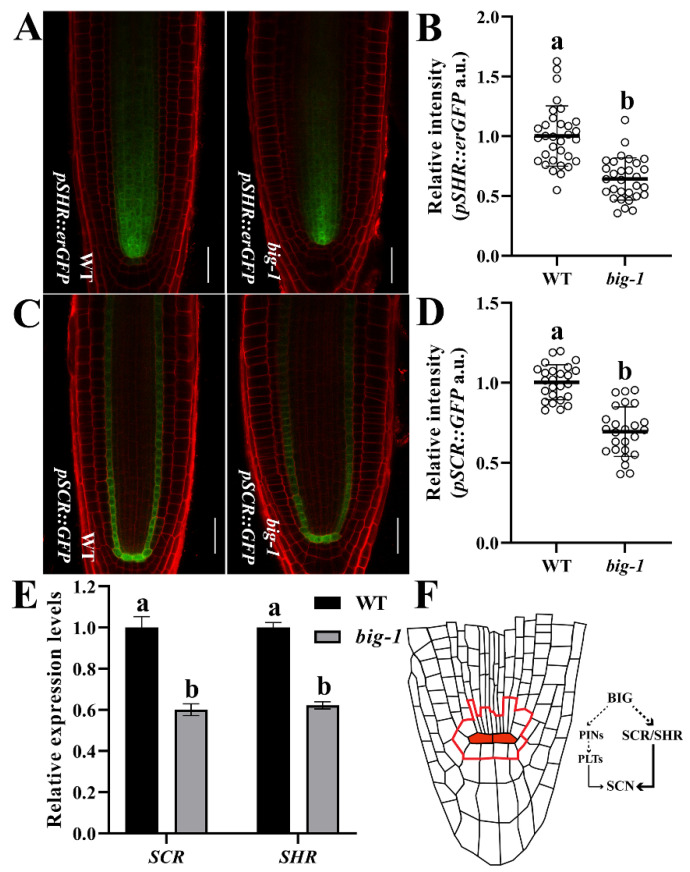
Mutations in *BIG* gene impact the expression of *SHR* and *SCR* genes. (**A**,**C**) *pSHR::erGFP* (**A**) and *pSCR::GFP* (**C**) expression in WT and *big-1* seedlings at 5 DAG. Scale bars, 20 µm. (**B**,**D**) Quantification of *pSHR::erGFP* (**B**) and *pSCR::GFP* (**D**) relative intensity in WT and *big-1* seedlings at 5 DAG. Values are means ± SD (*n* ≥ 30). Student’s *t*-test. Different letters indicate values are significantly different (*p* < 0.01). (**E**) qRT-PCR analyses show the relative expression levels of *SHR* and *SCR* in WT and *big-1* seedlings. Error bars represent SD. Two-way ANOVA, Tukey’s multiple comparisons test. Different letters indicate values are significantly different (*p* < 0.01). (**F**) Proposed mechanism of BIG in regulating the development of SCN.

## Data Availability

Not applicable.

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
