# Peer review of "BIG Modulates Stem Cell Niche and Meristem Development via SCR/SHR Pathway in Arabidopsis Roots"

_ijms, 2022, doi:10.3390/ijms23126784_

Round 1
Reviewer 1 Report
The data are well presented a well analyzed.
It is interesting to mention that several groups donated mutant and transgenic lines that facilitate importantly the development of this work.
Ad units in Figure 1F after 5,4,3.
BIG protein is related to proteasome activity? It is possible to use chemical inhibition or mutants with defects in proteasome activity to go further in the knowledge of BIG functions?
It should be appreciated a model including the contribution to the auxin transport and stem cell niche development obtained in this work and others work to clear the contribution of this investigation since there have been described several alleles of the BIG mutant.
Author Response
Referee: #1
- English language and style are fine/minor spell check required
Our response. We fully accept this point, and have had the manuscript carefully edited.
- Ad units in Figure 1F after 5,4,3.
Our response. We have added Ad units in Figure 1F after 5,4,3. Also updated the figures in the revised paper.
- BIG protein is related to proteasome activity? It is possible to use chemical inhibition or mutants with defects in proteasome activity to go further in the knowledge of BIG functions?
Our response. These suggestions were very useful. Actually, we have attempted to verify the E3 ubiquitin ligase activity of BIG by biochemical experiments, but failed. It is highly likely due to the extra-large size of the putative BIG protein (560KD) as it is extremely difficult to construct the BIG gene into a vector and express it successfully. Therefore, at present, we can only speculate that BIG protein might function as an E3 ubiquitin ligase based on the conserved domain of BIG protein. SCR is subjected to 26S-proteasome-mediated degradation, whereas SHR is not subjected to 26S-proteasome-mediated degradation (Cruz-Ramírez et al. 2012). Therefore, the effect of exogenous use of chemical inhibition to alter proteasome activity on SCR and SHR might be very complex.
- It should be appreciated a model including the contribution to the auxin transport and stem cell niche development obtained in this work and others work to clear the contribution of this investigation since there have been described several alleles of the BIG mutant.
Our response. Thanks for drawing this to our attention. We have updated the working model in the revised paper.

Reviewer 2 Report
The topic is interesting and the approach scientifically sound. Authors developed a better understanding regarding the specific mechanism of the BIG gene involved in root development. Figures are of good quality, indroduction is sufficient and methods and materials are explanatory. Authors are encouraged to pay some more effort in order to further heghlight the potential utility of their findings.
Author Response
Referee: #2
- English language and style are fine/minor spell check required
Our response. We fully accept this point, and have had the manuscript carefully edited.
